# Investigating Gender Bias in Language Models Using Causal Mediation Analysis

**Jesse Vig**[*1]       **Sebastian Gehrmann**[*2]       **Yonatan Belinkov**[*2]
**Sharon Qian**[2]       **Daniel Nevo**[3]       **Yaron Singer**[2]       **Stuart Shieber**[2]
[1] Salesforce Research          [2] Harvard University          [3] Tel Aviv University
jvig@salesforce.com          danielnevo@tauex.tau.ac.il
{gehrmann,belinkov,sharonqian,yaron,shieber}@seas.harvard.edu

## Abstract

Many interpretation methods for neural models in natural language processing investigate how information is encoded inside hidden representations. However, these methods can only measure whether the information exists, not whether it is actually used by the model. We propose a methodology grounded in the theory of causal mediation analysis for interpreting which parts of a model are causally implicated in its behavior. The approach enables us to analyze the mechanisms that facilitate the flow of information from input to output through various model components, known as mediators. As a case study, we apply this methodology to analyzing gender bias in pre-trained Transformer language models. We study the role of individual neurons and attention heads in mediating gender bias across three datasets designed to gauge a model's sensitivity to gender bias. Our mediation analysis reveals that gender bias effects are concentrated in specific components of the model that may exhibit highly specialized behavior.

## 1  Introduction

The success of neural network models in various natural language processing tasks, coupled with their opaque nature, has led to much interest in interpreting and analyzing such models. One goal of these analyses is to identify whether a model utilizes latent information in its internal representations to arrive at a prediction. This is of particular importance when diagnosing the reasons for a *biased* prediction. A popular class of analysis methods, often called structural analysis, aims to extract this information using probing classifiers that predict linguistic properties from representations of trained models (e.g., Adi et al., 2017; Conneau et al., 2018; Hupkes et al., 2018; Tenney et al., 2019). However, since probing classifiers only yield a correlational measure between a model's representations and an external property (Belinkov and Glass, 2019), they cannot show if the property is *causally* connected to the model's predictions. Moreover, Barrett et al. (2019) showed that probing classifiers may generate unfaithful interpretations and fail to generalize to unseen data.

We introduce a methodology for interpreting neural models to address these limitations. We adapt *causal mediation analysis* (Pearl, 2001) for analyzing the mechanisms by which information flows from input to output through different model components. Mediation analysis relies on measuring the change in an output following a counterfactual intervention in an intermediate variable, or *mediator*. Through such interventions, one can measure the degree to which inputs influence outputs directly (*direct effect*), or indirectly through the mediator (*indirect effect*). In our case, the mediator can be any model components that we wish to study, such as neurons or attention heads. We propose multiple controlled interventions in these mediators, which reveal their causal role in a model's behavior.

---

[*] Equal contribution. Y.B. is now at the Technion – Israel Institute of Technology. Work conducted while J.V. was at Palo Alto Research Center.

In a case study, we apply this framework to the analysis of gender bias in large pre-trained language models. Gender bias has surfaced as a major concern in word representations, with strong effects in both static word embeddings (Caliskan et al., 2017; Bolukbasi et al., 2016) and contextualized word representations (Zhao et al., 2019; Basta et al., 2019; Tan and Celis, 2019). Mediation analysis enables us to study how biased predictions arise from different model components. In our study, we focus on the role of individual neurons or attention heads in Transformer-based language models, in particular, several versions of GPT2 (Radford et al., 2019). In an experiment using several datasets designed to gauge a model's gender bias, we find that gender bias effects increase with larger models, which potentially absorb more bias from the underlying training data. The causal mediation analysis further reveals that gender bias is sparse, with much of the effect concentrated in a relatively small proportion of neurons and attention heads.

In summary, this paper makes two broad contributions. First, we cast causal mediation analysis as an approach for analyzing neural NLP models, which may be applied to a variety of models and phenomena. Second, we demonstrate this methodology in the case of analyzing gender bias in pre-trained language models, revealing the internal mechanisms by which bias effects flow from input to output through various model components. The code for reproducing our results is available at `https://github.com/sebastianGehrmann/CausalMediationAnalysis`.

## 2 Methodology

### 2.1 Preliminaries

Consider a large pre-trained neural language model (LM), parameterized by $\theta$, which predicts the probability of the next word given a prefix: $p_\theta(x_t \mid x_1, \ldots, x_{t-1})$. We will focus on LMs based on Transformers (Vaswani et al., 2017), although much of the methodology applies to other architectures as well. Let $\boldsymbol{h}_{l,i} \in \mathbb{R}^K$ denote the (contextual) representation of word $i$ in layer $l$ of the model, with neuron activations $\boldsymbol{h}_{l,i,k}$ ($1 \leq k \leq K$). These representations are composed using so-called multi-headed attention. Let $\alpha_{l,h,i,j} \geq 0$ denote the attention directed from word $i$ to word $j$ by head $h$ in layer $l$, such that $\sum_j \alpha_{l,h,i,j} = 1$.

### 2.2 Causal Mediation Analysis

Causal mediation analysis aims to measure how a treatment effect is mediated by intermediate variables (Robins and Greenland, 1992; Pearl, 2001; Robins, 2003). Pearl (2001) described an example where a side effect of a drug may cause patients to take aspirin, and the latter has a separate effect on the disease the drug was originally prescribed for. Thus, the drug has a direct effect through its standard mechanism and an indirect effect operating via aspirin taking.

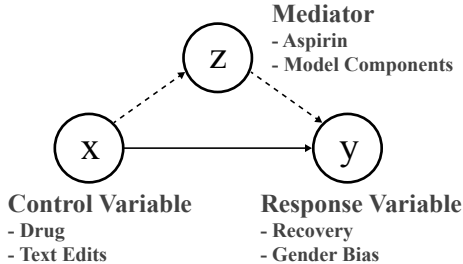

Figure 1: Mediation analysis illustration.

As illustrated in Figure 1, we may consider each neuron in a neural network to be analogous to aspirin in the example above – the neuron is influenced by the input and, in turn, affects the model output; however, there also exist direct pathways from the input to the output that do not pass through the neuron. We can thus decouple model components from the rest of the model by framing them as intermediaries in the causal path from inputs to outputs. Throughout this paper, we specifically focus on the use case of gender bias in language models, as past work suggests that gender is captured in specific model components, e.g., subspaces of contextual word representations (Zhao et al., 2019). By measuring the direct and indirect effects of targeted interventions, we can pinpoint how gender bias propagates through different parts of pre-trained LMs. While we use gender bias as a case study, the approach can be applied to other biases as well (race, ethnicity, etc.).

The example on the right illustrates a typical problem with biased LMs. Given a prompt $u$ such as *The nurse said that*, a language model generates a continuation. A biased model may assign a higher likelihood to *she* than to *he*, such that $p_\theta(she \mid u) > p_\theta(he \mid u)$. In this case, *she* is

**Prompt** $u$**:** The nurse said that __
**Stereotypical candidate:** she
**Anti-stereotypical candidate:** he

the stereotypical candidate, while *he* is the anti-stereotypical candidate, which reflects a societal bias associating nurses with women more than men. Coming back to the binary setup, the relative

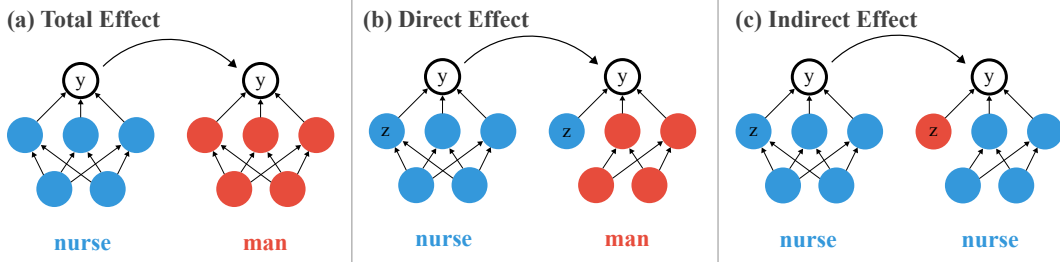

**(a) Total Effect**     **(b) Direct Effect**     **(c) Indirect Effect**

Figure 2: Mediation analysis illustration. Here the *do*-operation is $\boldsymbol{x} = \texttt{set-gender}$, which changes $\boldsymbol{u}$ from *nurse* to *man* in this example. The **total effect** measures the change in $\boldsymbol{y}$ resulting from the intervention; the **direct effect** measures the change in $\boldsymbol{y}$ resulting from performing the intervention while holding a mediator $\boldsymbol{z}$ fixed; the **indirect effect** measures the change caused by setting $\boldsymbol{z}$ to its value under the intervention, while holding $\boldsymbol{u}$ fixed.

probabilities assigned to the candidates can be thought of as a measure of gender bias in the model:

$$\boldsymbol{y}(u) = \frac{p_\theta(\text{anti-stereotypical} \mid u)}{p_\theta(\text{stereotypical} \mid u)}. \tag{1}$$

In our example, $\boldsymbol{y}(u) = p_\theta(he \mid \textit{The nurse said that})/p_\theta(she \mid \textit{The nurse said that})$. If $\boldsymbol{y}(u) < 1$, the prediction is stereotypical; if $\boldsymbol{y}(u) > 1$, it is anti-stereotypical. A perfectly unbiased model would achieve $\boldsymbol{y}(u) = 1$ and exhibit bias toward neither the stereotypical nor the anti-stereotypical case. This binary simplification of grammatical gender does not capture the full spectrum, as argued by Cao and Daumé III (2019), and it is not defined, for example, what probability mass a gender-neutral reference should receive[2]. While we leave extension of the framework to a continuous setup to future work, we report experimental results on the singular *they* compared to *he*.

We then apply causal mediation analysis by performing interventions on the input text, and measuring the effect on the gender bias measure defined above (Eq. 1), which we treat as the response variable. We define the following *do*-operations: (a) `set-gender`: replace the ambiguous profession with an anti-stereotypical gender-specific word (that is, replace *nurse* with *man*, *doctor* with *woman*, etc.); (b) `null`: leave the sentence as is. The population of units for this analysis is a set of example sentences such as the above prompt. We define $\boldsymbol{y}_x(u)$ as the value that $\boldsymbol{y}$ attains in unit $\boldsymbol{u} = u$ under the intervention $do(\boldsymbol{x} = x)$.

The unit-level **total effect** (TE) of $\boldsymbol{x} = x$ on $\boldsymbol{y}$ in unit $\boldsymbol{u} = u$ is the proportional difference[3] between the amount of bias under a gendered reading and under an ambiguous reading (Figure 2a):

$$\text{TE}(\texttt{set-gender, null}; \boldsymbol{y}, u) = \frac{\boldsymbol{y}_{\texttt{set-gender}}(u) - \boldsymbol{y}_{\texttt{null}}(u)}{\boldsymbol{y}_{\texttt{null}}(u)} = \frac{\boldsymbol{y}_{\texttt{set-gender}}(u)}{\boldsymbol{y}_{\texttt{null}}(u)} - 1. \tag{2}$$

For our running example, this results in

$$\frac{p_\theta(he \mid \textit{The man said that})}{p_\theta(she \mid \textit{The man said that})} \Big/ \frac{p_\theta(he \mid \textit{The nurse said that})}{p_\theta(she \mid \textit{The nurse said that})} - 1. \tag{3}$$

An illustrative example of the computation of the total effect is provided in Figure 3.

The average total effect of $\boldsymbol{x} = x$ on $\boldsymbol{y}$ is calculated by taking the expectation over the population $u$:

$$\text{TE}(\texttt{set-gender}, \texttt{null}; \boldsymbol{y}) = \mathbb{E}_u \left[ \boldsymbol{y}_{\texttt{set-gender}}(u)/\boldsymbol{y}_{\texttt{null}}(u) - 1 \right]. \tag{4}$$

We then analyze the causal role of specific mediators, or intermediary variables, which lie between $\boldsymbol{x}$ and $\boldsymbol{y}$. The mediator, denoted as $\boldsymbol{z}$, might be a particular neuron, a full layer, an attention head, or a certain attention weight. Following Pearl's definitions, we measure the direct and indirect effects of intervening in the model relative to $\boldsymbol{z}$ (Pearl, 2001).

Figure 3: An example calculation of the **total effect** with the prompt $u = $ *The nurse said that* and the control variable $x = \texttt{set-gender}$. Before the intervention, the model assigns a much higher probability to [she], the stereotypical example, than to [he]. By changing nurse to man, we compute the proportional probability of a definitionally gendered example. The total effect measures the effect of this intervention.

The **natural direct effect** (NDE) measures how much an intervention $x$ changes an outcome variable $\boldsymbol{y}$ directly, without passing through a hypothesized mediator $\boldsymbol{z}$. It is computed by applying the intervention $x$ but holding $\boldsymbol{z}$ fixed to its original value. For the present use case, we define the NDE of $\boldsymbol{x} = x$ on $\boldsymbol{y}$ given mediator $\boldsymbol{z} = z$ to be the change in the amount of bias when genderizing all units $u$, e.g., changing *nurse* to *man*, while holding $\boldsymbol{z}$ for each unit to its original value. This measures the direct effect on gender bias that does not pass through the mediator $\boldsymbol{z}$ (illustrated in Figure 2b):

$$\text{NDE}(\texttt{set-gender}, \texttt{null}; \boldsymbol{y}) = \mathbb{E}_u[\boldsymbol{y}_{\texttt{set-gender}, \boldsymbol{z}_{\text{null}}(u)}(u)/\boldsymbol{y}_{\text{null}}(u) - 1]. \tag{5}$$

The **natural indirect effect** (NIE) measures how much the intervention $x$ changes $\boldsymbol{y}$ indirectly, through $\boldsymbol{z}$. It is computed by setting $\boldsymbol{z}$ to its value under the intervention $x$, while keeping everything else to its original value. Thus the indirect effect captures the influence of a mediator on the outcome variable. For the present use case, we define the NIE as the change in amount of bias when keeping unit $u$ as is, but setting $\boldsymbol{z}$ to the value it would attain under a genderized reading. This measures the indirect effect flowing from $\boldsymbol{x}$ to $\boldsymbol{y}$ through $\boldsymbol{z}$ (Figure 2c):

$$\text{NIE}(\texttt{set-gender}, \texttt{null}; \boldsymbol{y}) = \mathbb{E}_u[\boldsymbol{y}_{\text{null}, \boldsymbol{z}_{\texttt{set-gender}}(u)}(u)/\boldsymbol{y}_{\text{null}}(u) - 1]. \tag{6}$$

This framework allows evaluating the causal contribution of different mediators $\boldsymbol{z}$ to gender bias. Through the distinction between direct and indirect effect, we can measure how much of the total effect of gender edits on gender bias flows through a specific component (indirect effect) or elsewhere in the model (direct effect). We experiment with mediators at the neuron level and the attention level.

## 2.3 Neuron Interventions

To study the role of individual neurons in mediating gender bias, we assign $\boldsymbol{z}$ to each neuron $\boldsymbol{h}_{l,\cdot,k}$ in the LM. The dataset we use consists of a list of templates that are instantiated by profession terms, resulting in examples such as *The nurse said that*. For each example, we define the $\texttt{set-gender}$ operation to move in the anti-stereotypical direction, changing female-stereotypical professions like *nurse* to *man* and male-stereotypical professions like *doctor* to *woman*. Section 3 provides more information on the dataset. We additionally investigate the effect of a gender-neutral intervention, for which we pick *person* as target of the $\texttt{set-gender}$ change and we measure the probability of the continuation *they*. Note that, unfortunately, all examples can be seen as biased against gender-neutrality since the models have had limited exposure to the singular *they*. Moreover, this case suffers from the additional confounder that the model could assign probability to the plural *they* if it does not refer to the profession.

In the experiments, we investigate the effect of intervening on each neuron independently, as well as on multiple neurons concurrently. That is, the mediator $\boldsymbol{z}$ may be a set of neurons. In all cases, the mediator is in the representation corresponding to the profession word, such as *nurse* in the example.

## 2.4 Attention Interventions

For studying attention behavior, we focus on the attention weights, which define relationships between words. The mediators $z$, in this case, are the attention heads $\alpha_{l,h}$, each of which defines a distinct attention mechanism.

To study their role, we align our intervention approach with two resources for assessing gender bias in pronoun resolution: Winobias (Zhao et al., 2018a) and Winogender (Rudinger et al., 2018). Both datasets consist of Winograd-schema-style examples that aim to assess gender bias in coreference resolution systems. We reformulate the examples to study bias in LMs, as shown in the example on the right, taken from Winobias. According to the stereotypical reading, the pronoun *she* refers to the nurse, implying the continuation *was caring*. The anti-stereotypical reading links *she* to the farmer, this time implying the continuation *was screaming*. The bias measure is $y(u) = p_\theta(was\ screaming \mid u)/p_\theta(was\ caring \mid u)$.[4] In this case, we define the swap-gender operation, which changes *she* to *he*. The total effect is

> **Prompt** $u$: The nurse examined the farmer for injuries because she ____
> **Stereotypical candidate:** was caring
> **Anti-stereotypical candidate:** was screaming

$$\text{TE(swap-gender, null}; y, u) = y_{\text{swap-gender}}(u)/y_{\text{null}}(u) - 1. \qquad (7)$$

In the experiments, we study the effect of the attention from the last word (*she* or *he*) to the rest of the sentence.[5] Intuitively, in the above example, if the word *she* attends more to *nurse* than to *farmer*, then the more likely continuation might be *was caring*. We compute the NDE and NIE for each head individually by intervening on the attention weights $\alpha_{l,h,\cdot,\cdot}$. We also evaluate the joint effects when intervening on multiple attention heads concurrently. The population-level TE and the NDE and NIE are defined analogously as above.

## 3 Experimental Details

**Models** As an example large pre-trained LM, we use GPT2 (Radford et al., 2019), a Transformer-based (English) LM trained on massive amounts of data. We use several model sizes made available by Wolf et al. (2019): small, medium, large, extra-large (xl), and a distilled model (Sanh et al., 2019).

**Data** For neuron intervention experiments, we augment the list of templates from Lu et al. (2018) with several other templates, instantiated with professions from Bolukbasi et al. (2016). The templates have the form "The [occupation] [verb] because".[6] The professions are accompanied by crowdsourced ratings between $-1$ and $1$ for definitionality and stereotypicality. *Actress* is definitionally female, while *nurse* is stereotypically female. None of the professions are stereotypically or definitionally gender-neutral in the sense that those people working in the profession are referred to in singular *they*. To simplify processing by GPT2 and focus on common professions, we only use examples that are not split into sub-word units, resulting in 17 templates and 169 professions, 2,873 examples in total. The full lists of templates and professions are given in Appendix A.1. We refer to these examples as the Professions dataset.

For attention intervention experiments, we use examples from Winobias Dev/Test (Zhao et al., 2018a) and Winogender (Rudinger et al., 2018), totaling 160/130 and 44 examples that fit our formulation, respectively. We experiment with the full datasets and filtering by total effect. Both datasets include statistics from the U.S. Bureau of Labor Statistics to assess the gender stereotypicality of the referenced occupations. Appendix A.2 provides additional details about the datasets and preprocessing methods.

## 4 Results

### 4.1 Total Effects

Before describing the results from the mediation analysis, we summarize some insights from measurements of the total effect. Unless noted otherwise, the reported results stem from

Table 1: Total effects (TE) of gender bias in various GPT2 variants.

| Dataset | GPT2 variants | | | | | |
| | small rand. | distil | small | medium | large | xl |
| --- | --- | --- | --- | --- | --- | --- |
| Winobias | 0.066 | 0.118 | 0.249 | 0.774 | 0.751 | 1.049 |
| Winogender | 0.045 | 0.081 | 0.103 | 0.322 | 0.364 | 0.342 |
| Professions | 0.117 | 130.859 | 112.275 | 115.945 | 96.859 | 225.217 |

the binary male→female or female→male interventions. We report separately the results of male/female→neutral interventions, which due to their potentially confounded nature cannot be grouped with the rest of the results. Table 1 shows the total effects of gender bias in the different GPT2 models, on three datasets, as well as the effects with a randomly initialized GPT2-small model. Random model effects are much smaller, indicating that it is the training process that causes gender bias.

**Larger models are more sensitive to gender bias** In the Winograd-style datasets, the total effect mostly increases with model size, saturating at the large and xl models. In the professions dataset, model size is not well correlated with total effect, but GPT2-xl has a much larger effect. Since larger models can more accurately emulate the training corpus, it makes sense that they would more strongly integrate its biases.

**Effects in different datasets** It is difficult to compare effect magnitudes in the three datasets because of their different nature. The professions dataset yields much stronger effects than the Winograd-style datasets. This may be attributed to the more explicit source of bias, the word representations, as compared to intricate coreference relations in the Winograd-style datasets.

**Some effects are correlated with external gender statistics** In the professions dataset, we found moderate positive correlations between external gender bias[7] and the log-total effect, ranging from $0.35$ to $0.45$ over different models, indicating that the model captures the expected biases. It further shows that the effect is amplified by the model for words that are perceived as more biased. In the Winograd-style datasets, we found relatively low correlations between the log-total effect and the log-ratio of the two occupations' stereotypicality, ranging from $0.17$ to $0.26$. This low correlation may be due to a smaller size than the professions dataset or the more complex Winograd-style relations.

**The gender-neutral case leads to more consistent effects** In the neutral case, the baseline probability $p(they|u)$ is much more consistent, but low, across all professions. Consider the template "The X said that" — in this case, under GPT2-distil "they" varies in probability from 0.2% to 4.2% while "he" has a much wider range from 1.1% to 31.8%. Consequently, the total effect for neutral interventions is much more consistent across models and templates. GPT2-distill, GPT2-small, and GPT2-medium have total effects of 8.3, 7.5, and 9.6 respectively, all with standard deviations $< 10$, in the professions dataset. We hypothesize that this can mostly be attributed to very low probability for the singular "they" and a consistent baseline probability where "they" is part of a referential statement toward a group of individuals, for example in "The accountant said that they [the people] need to pay taxes".

## 4.2 Mediation Analysis

Where in the model are gender bias effects captured? Are the effects mediated by only a few model components or distributed across the model? Here we answer these questions by measuring the indirect effect flowing through different mediators.

**Attention** Figure 4a shows the indirect effects for each head in GPT2-small on Winobias. The heatmap shows interventions on each head individually. A small number of heads, concentrated in the middle layers of the model, have much higher indirect effects than others. The bar chart shows indirect effects when intervening on all heads in a single layer concurrently. Consistent with the head-level heatmap, the effects are concentrated in the middle layers. We found this sparsity consistent in all model variants and datasets we examined. We did not find similar behavior in a

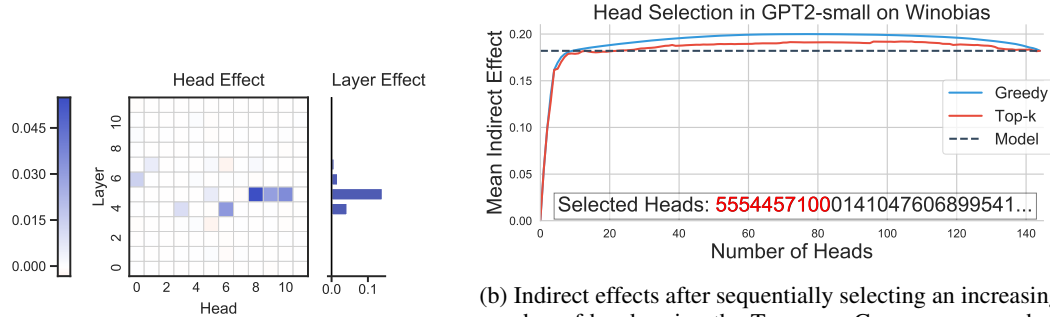

(a) Indirect effects in GPT2-small on Winobias for heads (the heatmap) and layers (the bar chart).

(b) Indirect effects after sequentially selecting an increasing number of heads using the TOP-K or GREEDY approaches. Very few heads are required to saturate the model effect. The inset lists the sequence of layers of heads selected by GREEDY. The ones in red together reach the model effect, demonstrating the concentration of the effect in layers 4 and 5.

Figure 4: Sparsity effects in attention heads.

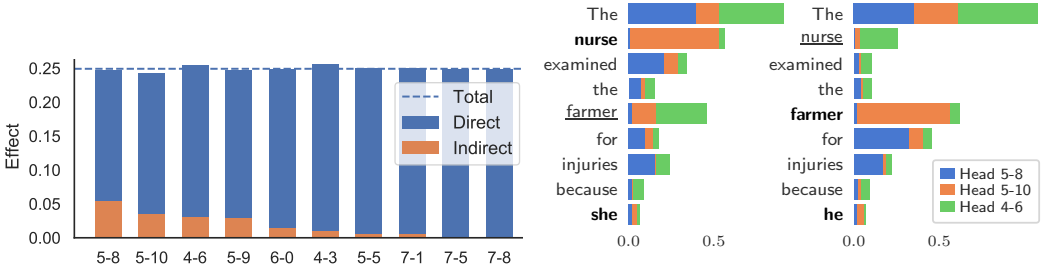

Figure 5: Top 10 heads by indirect effect in GPT2-small on Winobias, and their respective direct effects. Both effects appear largely additive with respect to total effect, a surprising result given the nonlinear nature of these models.

Figure 6: Attention in GPT2-small on a Winobias example, directed from either *she* or *he*. Head 5-10 attends directly to the **bold** stereotypical candidate, head 5-8 attends to the words following it, and head 4-6 attends to the underlined anti-stereotypical candidate. Attention to the first token may be null attention (Vig and Belinkov, 2019). Appendix C.2 shows more examples.

randomly initialized model, indicating that these patterns do not occur by chance. Appendix C.1 provides additional visualizations of indirect effects as well as direct effects.

The indirect and direct effects of the top attention heads are summarized in Figure 5. The total effect roughly equals the sum of the direct and indirect effects.[8] Qualitative analysis suggests that these top heads take on specialized roles with respect to gender bias, as illustrated in Figure 6. The figure demonstrates that attention heads capture different coreference aspects: one head aligns with the stereotypical coreference candidate, another head attends to the tokens following that candidate, while a third attends to the anti-stereotypical candidate. Vig (2019) previously identified the same head as relating to coreference resolution based on visual inspection and Clark et al. (2019) found an attention head in BERT (Devlin et al., 2019) that was highly predictive of coreference, also in layer 5 out of 12. The specialization of attention heads seems to be a general property of Transformers (Voita et al., 2019) and has been observed for a range of syntactic dependency relationships (Clark et al., 2019; Htut et al., 2019; Vig and Belinkov, 2019).

To determine how many heads are required to achieve the full effect of intervening on all heads, we also intervene on groups of heads. While the computational complexity of selecting a single head scales linearly with the number of total heads, selecting a group of heads scales polynomially and becomes computationally intractable. To efficiently select a subset of $k$ heads given $n$ total heads, we use two methods: a GREEDY approach, which iteratively selects the head with the maximal marginal contribution to the indirect effect and requires $O(nk)$ evaluations, and a TOP-K approach,

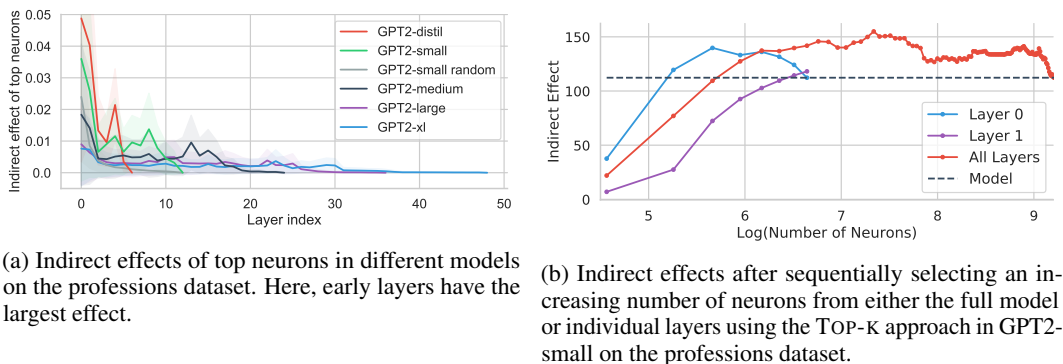

(a) Indirect effects of top neurons in different models on the professions dataset. Here, early layers have the largest effect.

(b) Indirect effects after sequentially selecting an increasing number of neurons from either the full model or individual layers using the TOP-K approach in GPT2-small on the professions dataset.

Figure 7: Sparsity effects in neurons.

which selects the $k$ elements with the strongest individual effects and requires $O(n)$ evaluations. Appendix D provides more information on these algorithms. Only 10 heads are required to match the effect of intervening on all 144 heads at the same time (Figure 4b). The first six selected ones are from layers 4 and 5, further demonstrating the concentration of the effect in the middle layers.

**Neurons**

Figure 7a shows the indirect effects from the top 5% of neurons from each layer in different models. The word embeddings (layer 0) and the first hidden layer have the strongest effects. This stands in contrast to the attention intervention results, where middle layers had much larger effects. However, we still observe a small increase in effect within the intermediate layers across all models except for the randomized one. Interestingly, we do not observe the same concentration for neutral intervention. As can be seen in Figure 8, where, for simplicity, we focus on GPT2-medium, the effects are distributed across all layers, but similarly increasing a bit toward the later middle layers.

Figure 7b shows the indirect effects when selecting neurons by the TOP-K algorithm.[9] Similar to the attention result, a tiny fraction of neurons (4%) is sufficient for obtaining an effect equal to that of intervening on all neurons concurrently. Most of the top selected neurons are concentrated in the embedding layer and first hidden layer. We show in Figure 8 that the same effect does not occur in gender-neutral interventions. The 100 neurons with the highest average indirect effect for gender-neutral interventions in GPT2-small appear within the embeddings and the first 9 of the 12 layers, while only about 30 of those come from the embedding

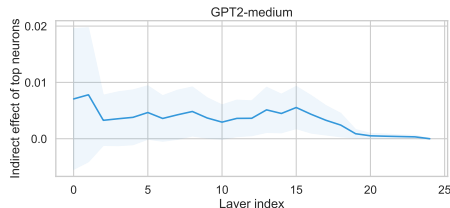

Figure 8: Indirect effects of top neurons in GPT2-medium for gender-neutral interventions on the professions dataset. Here, the effect is distributed across all layers.

and first layer. This finding, which is consistent across all model sizes, provides further evidence for the the lack of representation of gender-neutral information in embeddings.

## 5 Related Work

### 5.1 Analysis Methods

Methods for interpreting neural network models in NLP can be broadly divided into two types: *structural* and *behavioral*. Structural methods focus on identifying what information is contained in different model components. Probing classifiers aim to answer such questions by using models' representations as input to classifiers that predict various properties (Adi et al., 2017; Hupkes et al., 2018; Conneau et al., 2018). However, this approach is not connected to the model's behavior (that is, its predictions) on the task it was trained on (Belinkov and Glass, 2019; Tenney et al., 2019). The representation may thus have some information by coincidence, or by virtue of a shared cause, without it being used by the original model. In addition, it is challenging to differentiate the

information learned by the probing classifier from that learned by the underlying model (Hewitt and Liang, 2019). Behavioral approaches, on the other hand, assess how well a model captures different linguistic phenomena by evaluating the model's performance on curated examples (e.g., Sennrich, 2017; Isabelle et al., 2017; Naik et al., 2018). These methods directly evaluate a model's prediction but fail to provide insight into its internal structure. Another approach identifies important input features that contribute to a model's prediction via saliency methods (Li et al., 2016; Arras et al., 2017; Murdoch et al., 2018), which also typically ignore the model's internal structure, although one could compute them with respect to internal representations.

Our causal mediation analysis approach bridges the gap between these two lines of work, providing an analysis that is both structural and behavioral. Mediation analysis is an unexplored formulation in the context of interpreting deep NLP models. In recent work, Zhao and Hastie (2019) used mediation analysis for interpreting black-box models. However, their analysis was limited to simple datasets and models, while we focus on deep language models. Furthermore, they only considered total effects and (controlled) direct effects, while we measure (natural) direct and indirect effects, which is crucial for studying the role of internal model components.

## 5.2 Gender Bias and Other Biases

Neural networks learn to replicate historical, societal biases from training data in various tasks such as natural language inference (Rudinger et al., 2017), coreference resolution (Cao and Daumé III, 2019), and sentiment analysis (Kiritchenko and Mohammad, 2018). This conflicts with the principle of counterfactual fairness, which states that the model predictions should not be influenced by changes to a sensitive attribute such as gender (Kusner et al., 2017); for instance, a fair and unbiased model should equally associate gendered pronouns with professions. However, biased models make this association proportionally to the distribution of gender in the training data (Caliskan et al., 2017). While efforts have been made to reduce bias, this remains a significant ethical challenge.

A common strategy to mitigate biases is to change the training data (e.g., Lu et al., 2018; Hall Maudslay et al., 2019; Zhao et al., 2018a; Kaushik et al., 2019), the training process (e.g., Huang et al., 2019; Qian et al., 2019), or the model itself (e.g., Madras et al., 2019; Romanov et al., 2019; Gehrmann et al., 2019) to ensure counterfactual fairness. The resulting biases are often measured similarly to this work by testing that mentions of occupations lead to equal probabilities across grammatical genders in referential expressions. Others have focused on de-biasing word embeddings and contextual word representations (Bolukbasi et al., 2016; Zhao et al., 2018b; Yang and Feng, 2020), though recent work has questioned the efficacy of these debiasing techniques (Elazar and Goldberg, 2018; Gonen and Goldberg, 2019).

Our work contributes a novel perspective to the literature on gender bias in neural NLP model by characterizing the role of mediators in biased predictions via performing interventions.

## 6 Discussion and Conclusion

This paper introduced a framework for interpreting neural NLP models based on causal mediation analysis. An application of this framework yields several insights regarding the mechanisms by which gender bias is mediated in Transformer LMs. We find that larger models have a greater capacity to absorb gender bias, though this bias manifests in a relatively small proportion of neurons and attention heads. Qualitative analysis suggests that model components may take on specialized roles in propagating gender bias.

This framework can be extended in multiple ways, for example to work with different model architectures and to analyze different and potentially continuous or multi-class biases (Cao and Daumé III, 2019). The results could also be applied to control model outputs to generate text with fair representation (Giulianelli et al., 2018; Dathathri et al., 2020). The causality literature offers many avenues for continuing this line of work, including mediation analysis with non-linear models, and alternative effect decompositions (Imai et al., 2010a,b; VanderWeele and Vansteelandt, 2009). A promising direction is to focus on path-specific effects (Avin et al., 2005), to identify the exact mechanisms through which biases arise. Characterizing specific paths from model input to output might also be useful during training by disincentivizing the creation of paths leading to bias.

## Broader Impact

This work focuses on the identification and analysis of biases that large language models acquire during training. Following the reasoning of Rawls (1958) among others, it is impermissible to use models that treat persons, groups, or institutions differently based on their attributes. Yet, language models are widely applied in real world settings. To remedy the effect of the implicit discrimination that this may cause, it is imperative to develop unbiased models. Understanding the causal mechanisms within neural networks is critical to developing trustworthy and provably fair models. Our method presents a first step toward the active debiasing of such models, as discussed in Section 6. Moreover, since model biases mirror societal biases, as shown by Caliskan et al. (2017) and confirmed in Section 4.1, our method may be of interest to those studying these biases in large corpora.

However, while our case study presents a best effort to cover different cases and linguistic phenomena, it is not possible to fully cover all cases of gender bias within a language using only a limited set of constructed templates. Importantly, the main focus of our study uses a limited binary setup, which does not easily lend itself to applications on grammatical gender. We tried to avoid implicit discrimination of individuals who do not identify as either male or female by conducting experiments on a gender-neutral pronoun, but more work needs to be done on generating inclusive referring expressions that cover all possible pronouns. Moreover, since the model confuses the singular for the plural *they*, it will require additional disambiguation efforts to apply our methodology in this case. Finally, this study focuses only on the English language. The conclusions drawn from our results may not generalize to other languages or linguistic phenomena. In this case, our findings may lead researchers down the wrong path.

## Acknowledgments

S. G. was supported by a Siebel Fellowship. Y.B. was supported by the Harvard Mind, Brain, and Behavior Initiative. Work conducted while J.V. was at Palo Alto Research Center. S.Q. and Y.S. were supported by BSF grant 2014389, NSF grant CAREER CCF-1452961, NSF USICCS proposal 1540428, Google research award, and a Facebook research award.

## Footnotes

[2]One could argue for the case where *he*, *she*, and *they* have same probability or where *they* has equal representation to the sum of *he* and *she*. Alternatively, one could argue that grammatical genders are inherently discriminatory and that we should change all of them to *they*, unless we know an individual's preferred pronouns.

[3]We make the difference proportional to control for the high variance of $\boldsymbol{y}$ across examples (appendix A.1).

[4]To compute probabilities of multi-word continuations, we use the geometric mean of the token probabilities.

[5]One may also study individual attention arcs. However, attention does not always focus on a specific word, often falling on adjacent words. See Appendix C.2 for this phenomenon.

[6]The original list only includes examples ending with *because*. To increase the lexical diversity of examples, we add templates with other conjunctions

[7]For this analysis, we add each profession's stereotypicality and definitionality as the overall bias value.

[8]This kind of decomposition is expected in a linear model, but may be surprising in a non-linear model like GPT2. Still, we found it consistent in all our analyses, so we focus on showing the indirect effect results.

[9]For computational reasons, we select sets of 96 neurons.

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
