[Supplementary Material]

The <occupation> said that . . .
The <occupation> yelled that . . .
The <occupation> whispered that . . .
The <occupation> wanted that . . .
The <occupation> desired that . . .
The <occupation> wished that . . .
The <occupation> ate because . . .
The <occupation> ran because . . .
The <occupation> drove because . . .
The <occupation> slept because . . .
The <occupation> cried because . . .
The <occupation> laughed because . . .
The <occupation> went home because . . .
The <occupation> stayed up because . . .
The <occupation> was fired because . . .
The <occupation> was promoted because . . .
The <occupation> yelled because . . .

Figure 9: Templates for neuron interventions.

# A  Data Preparation

## A.1  Professions Dataset

Figure 9 shows the 17 base templates used for the neuron interventions. To validate that each template would capture gender bias, we instantiate each with an occupation of *woman* and *man* and verify that the conditional probabilities of *she* and *he* align with gender. Given *woman* as the occupation word, the probability ratio $p(\text{she})/p(\text{he})$ ranges from 2.5 to 45.1 across templates ($\mu = 17.2, \sigma = 13.1$). Given *man*, the value $p(\text{he})/p(\text{she})$ ranges from 3.0 to 55.4 ($\mu = 21.9, \sigma = 16.2$). Thus the relative probabilities align with gender, though they vary greatly in magnitude.

For each of the templates, we used the following professions. Words in *italics* are definitional and were thus excluded from the total effect calculation:

**female:** *actress*, advocate, aide, artist, baker, clerk, counselor, dancer, educator, instructor, maid, *nun*, nurse, observer, performer, photographer, planner, poet, protester, psychiatrist, secretary, singer, substitute, teacher, teenager, therapist, treasurer, tutor, *waitress*
**neutral:** acquaintance, character, citizen, correspondent, employee, musician, novelist, psychologist, student, writer
**male:** accountant, *actor*, administrator, adventurer, ambassador, analyst, architect, assassin, astronaut, astronomer, athlete, attorney, author, banker, bartender, biologist, bishop, boss, boxer, broadcaster, broker, *businessman*, butcher, campaigner, captain, chancellor, chef, chemist, cleric, coach, collector, colonel, columnist, comedian, comic, commander, commentator, commissioner, composer, conductor, congressman, consultant, cop, critic, curator, *dad*, dean, dentist, deputy, detective, diplomat, director, doctor, drummer, economist, editor, entrepreneur, envoy, farmer, filmmaker, firefighter, *fisherman*, footballer, goalkeeper, guitarist, historian, inspector, inventor, investigator, journalist, judge, landlord, lawmaker, lawyer, lecturer, legislator, lieutenant, magician, magistrate, manager, mathematician, mechanic, medic, midfielder, minister, missionary, *monk*, narrator, negotiator, officer, painter, pastor, philosopher, physician, physicist, *policeman*, politician, preacher, president, priest, principal, prisoner, professor, programmer, promoter, prosecutor, protagonist, rabbi, ranger, researcher, sailor, saint, *salesman*, scholar, scientist, senator, sergeant, servant, soldier, solicitor, strategist, superintendent, surgeon, technician, trader, trooper, *waiter*, warrior, worker, wrestler

## A.2  Winobias and Winogender

For both Winobias and Winogender datasets, we exclude templates in which the shared prompt does not end in a pronoun.[10] For Winobias, we only consider *Type 1* examples, which follow the format of a shared prompt and two alternate continuations. We also experiment with filtering by total effect,

Table 2: Number of examples from Winobias and Winogender datasets, including filtered (Filt.) and unfiltered (Unfilt.) versions. The size of the filtered versions vary between models because each model produces different total effects (used for the filtering). The number of examples excluded due to format (not included in the above numbers) were 38, 68, and 16 for Winobias Dev, Winobias Test, and Winogender, respectively.

| | Winobias | | | | Winogender | | | |
| | Dev | | Test | | BLS | | Bergsma | |
| Model | Filt. | Unfilt. | Filt. | Unfilt. | Filt. | Unfilt. | Filt. | Unfilt. |
| --- | --- | --- | --- | --- | --- | --- | --- | --- |
| GPT2-distil | 61 | 160 | 51 | 130 | 15 | 44 | 18 | 44 |
| GPT2-small | 87 | 160 | 66 | 130 | 21 | 44 | 20 | 44 |
| GPT2-medium | 99 | 160 | 79 | 130 | 23 | 44 | 27 | 44 |
| GPT2-large | 94 | 160 | 69 | 130 | 24 | 44 | 26 | 44 |
| GPT2-xl | 101 | 160 | 72 | 130 | 25 | 44 | 26 | 44 |

Table 3: Total effects on Winobias and Winogender, including filtered (Filt.) and unfiltered (Unfilt.) versions.

| | Winobias | | | | Winogender | | | |
| | Dev | | Test | | BLS | | Bergsma | |
| Model | Filt. | Unfilt. | Filt. | Unfilt. | Filt. | Unfilt. | Filt. | Unfilt. |
| --- | --- | --- | --- | --- | --- | --- | --- | --- |
| GPT2-distil | 0.118 | 0.012 | 0.127 | 0.023 | 0.081 | 0.005 | 0.075 | 0.011 |
| GPT2-small | 0.249 | 0.115 | 0.225 | 0.098 | 0.103 | 0.020 | 0.135 | 0.040 |
| GPT2-medium | 0.774 | 0.474 | 0.514 | 0.311 | 0.322 | 0.128 | 0.384 | 0.231 |
| GPT2-large | 0.751 | 0.427 | 0.492 | 0.238 | 0.364 | 0.173 | 0.350 | 0.192 |
| GPT2-xl | 1.049 | 0.660 | 0.754 | 0.400 | 0.342 | 0.168 | 0.362 | 0.202 |

removing examples with a negative total effect as well as examples in the bottom quartile of those with a positive total effect. The sizes of all dataset variations may be found in Table 2. Results are reported for filtered versions of both datasets and the Dev set of Winobias unless otherwise noted.

Both datasets include statistics from the U.S. Bureau of Labor Statistics (BLS) to assess the gender stereotypicality of the referenced occupations. Winogender additionally includes gender estimates from text (Bergsma and Lin, 2006), which we also include in our analysis. Whereas each Winobias example includes two occupations of opposite stereotypicality, each Winogender example includes one occupation and a *participant*, for which no gender statistics are provided. For consistency with the Winobias analysis, we make the simplifying assumption that the gender stereotypicality of the participant is the opposite of that of the occupation.

## B Additional Total Effects

Table 3 provides the total effects across all variations of the Winograd-style datasets. The relationship between model and effect size is relatively consistent across dataset variations (Winobias/Winogender, filtered/unfiltered, Dev/Test, BLS/Bergsma gender statistics), though the magnitudes of the effects may vary between dataset variations.

Table 4 provides the total effects on the professions dataset when separated to stereotypically female and male professions, where stereotypicality is defined by the profession statistics provided by Bolukbasi et al. (2016). Notably, the effects are much larger in the female case. This may be explained by stereotypicaly-female professions being of higher stereotypicality than stereotypically-male professions, reflecting a societal bias viewing women's professions as more narrowed.

Table 4: Total effects (TE) of gender bias in various GPT2 variants evaluated on the professions dataset, when separating by gender-stereotypicality.

| Model | Female | Male | All |
|---|---|---|---|
| GPT2-small rand. | 0.10 | 0.19 | 0.12 |
| GPT2-distil | 155.31 | 23.47 | 130.86 |
| GPT2-small | 129.36 | 15.16 | 112.28 |
| GPT2-medium | 120.60 | 94.75 | 115.95 |
| GPT2-large | 107.44 | 48.99 | 96.86 |
| GPT2-xl | 255.22 | 89.31 | 225.22 |

(a) GPT2-distil

(b) GPT2-medium

(c) GPT2-large

(d) GPT2-xl

Figure 10: Mean indirect effect on Winobias for heads (the heatmap) and layers (the bar chart) over additional GPT2 variants.

## C   Additional Attention Results

### C.1   Indirect and Direct Effects

Figure 10 complements Figure 4a by visualizing the indirect effects for additional GPT2 models. As with Figure 4a, the attention heads with the largest indirect effects lie in the middle layers of each model. Figure 11 shows the indirect effects for a model with randomized weights. Figures 12 and 13 visualize the indirect effects for other dataset variations for the GPT2-small model from Figure 4a. The attention heads with largest indirect effect have significant overlap across the dataset variations.

Figure 14 visualizes *direct* effects on Winobias for GPT2-small and GPT2-large. As discussed in Section 4.2, the sum of direct and indirect effects approximate the total effect.

Figure 11: Indirect effect when using a randomly initialized GPT2-small model on Winobias.

(a) Filtered, Dev

(b) Unfiltered, Dev

(c) Filtered, Test

(d) Unfiltered, Test

Figure 12: Indirect effect for Winobias (GPT2-small).

(a) Filtered, BLS

(b) Unfiltered, BLS

(c) Filtered, Bergsma

(d) Unfiltered, Bergsma

Figure 13: Indirect effect for Winogender (GPT2-small).

(a) GPT2-small

(b) GPT2-large

Figure 14: Direct effect for Winobias for GPT2-small and GPT2-large.

Figure 15: Attention of different heads across the ten Winobias examples with greatest total effect for the GPT2-small model. The stereotypical candidate is in **bold** and the anti-stereotypical candidate is underlined. Attention roughly follows the pattern described in Figure 6.

## C.2 Examples

Figure 15 visualizes attention for the Winobias examples with the greatest total effect in GPT2-small, complementing the example shown in Figure 6. Figure 16 visualizes attention for additional models for the same example shown in Figure 6.

(a) Attention for GPT2-distil. Most attention is directed to the first token (null attention). Head 3-1 attends primarily to the **bold** stereotypical candidate, head 2-6 attends to the underlined anti-stereotypical candidate, and attention from head 3-6 is roughly evenly distributed.

(b) Attention for GPT2-medium. Head 10-12 attends directly to the **bold** stereotypical candidate, and heads 10-9 and 6-15 attend to the following words.

(c) Attention for GPT2-large. Heads 16-5 and 15-6 attend to the **bold** stereotypical candidate and optionally the following word. Head 16-19 attends to the words following the underlined anti-stereotypical candidate.

(d) Attention for GPT2-xl. Heads 16-5 and 17-10 attend primarily to the word following the **bold** stereotypical candidate. Head 16-24 attends primarily to the words following the underlined anti-stereotypical candidate.

Figure 16: Attention of top 3 heads on an example from Winobias, directed from either *she* or *he*, across different GPT2 models. The colors correspond to different heads. The results for GPT2-small are shown in Figure 6.

# D   Additional subset selection results

We wish to select a subset of attention heads or neurons that perform well together to better understand the sparsity of attention heads and neurons and their impact on gender bias in Transformer models.

The problem of subset selection (selecting $k$ elements from $n$) is an NP-hard combinatorial optimization problem. To construct a meaningful solution set, we employ several algorithms for subset selection from submodular maximization. We note that while our objective functions are not strictly submodular as they do not satisfy the diminishing returns property, our objectives exhibit submodular-like properties and numerous algorithms have been proposed to efficiently maximize submodular and variants of submodular functions.

For monotone submodular functions, it is known that a greedy algorithm that iteratively selects the element with the maximal marginal contribution to its current solution obtains a $1-1/e$ approximation for maximization under a cardinality constraint (Nemhauser and Wolsey, 1978) and that this bound is optimal. For non-monotone submodular functions, there is the randomized greedy algorithm which emits a $1/e$ approximation to the optimal solution (Buchbinder et al., 2014).

To select subsets of attention heads, we compare TOP-K (selecting $k$ elements with the largest individual values) and GREEDY. Even though randomized greedy has stronger theoretical guarantees because our objective is clearly non-monotonic, we favor the deterministic algorithm for increased interpretability. Figure 17 shows results for head selection across different models on Winogender and Winobias. Sparsity is consistent across all experiments where only a small proportion of heads are sufficient to achieve the full model effect of intervening at all heads. On Winogender, only 4/4/5/4% of heads are needed to saturate, while on Winobias, only 6/7/8/6% of heads are needed in GPT2-distil/small/medium/large.

To select subsets of neurons, we use TOP-K to compute NIE of sets of neurons because sequential greedy is too computationally intensive to run. Alternative methods using adaptive sampling techniques have been proposed to speed-up GREEDY for submodular functions under cardinality constraints (Ene and Nguyen, 2019; Fahrbach et al., 2019b; Balkanski and Singer, 2018a,b). For non-monotone or non-submodular functions, there are parallelized algorithms that use similar techniques to select sets (Balkanski et al., 2018; Qian and Singer, 2019; Fahrbach et al., 2019a). These methods provide an alternative approach to TOP-K for selecting subsets of neurons and can be explored in future work.

Figure 17: The effect after sequentially selecting an increasing number of heads through the Top-k or Greedy approach on different model types and data. A small proportion of heads are required to saturate the effect of the model.

## Footnotes

[10] An example of a removed template is: "The receptionist welcomed the lawyer because *this is part of her job.*" / "The receptionist welcomed the lawyer because *it is his first day to work.*"