[Reviews · NeurIPS 2020]

Review 1

Summary and Contributions: Edit: I appreciate the thoughtful response from the authors. Based on author response, I am updating my score as most of my concerns have been addressed. I believe following suggested revisions, this paper will be a much stronger work in this field and I recommend a clear acceptance. The paper discusses an important problem with interpretability methods looking at how information is encoded in a neural model: most of them can identify what information is present but not whether a model uses it. This paper presents an interesting approach to solve this problem built on the causal theory of mediation. To test their approach, the authors select problems from NLP by focusing on the Professions, Winobias, and Winogender datasets and using the GPT-2 model for all experiments. In their approach, they consider individual (or collection of) neurons as mediators in a causal model wherein input text is causing the output label. The authors first calculate the treatment effect by intervening on text (applying the do-operator) and setting the gender attribute to a particular value (changing it from the stereotypical gender value to anti-stereotypical value, assuming gender to be binary). Then, for each mediator, they decompose this treatment effect into natural direct effect (that can be accounted for without the presence of the mediator) and natural indirect effect (that could only be justified by the mediator).

Strengths: This paper addresses a very important problem associated with interpreting how neural models encode information using a theoretically grounded mechanism. Results presented in the paper reflect the effectiveness of the proposed approach that is a principled way to identify how neural models encode (and use) information. This contribution is timely and relevant to the broader ML and NLP communities and I'm eager to try their approach on other tasks. While I find the results very interesting, I also have some concerns (laid below).

Weaknesses: The authors have not specified the assumptions that need to be made while conducting mediation analysis. The reason why this is even important here as the paper introducing GPT-2 architecture does not discuss the model architecture, it is hard to verify whether certain assumptions will hold or not. Absent certain assumptions, the findings in this paper may not hold. For instance, the causal mediation analysis requires an assumption that there is no unmeasured confounding on the M->Y or X->M paths in a simple model where X causes Y and M is the mediator which may not be true in certain cases (presence of residual or highway connections could violate that assumption). Other assumptions (such as consistency, SUTVA, etc. for those new to this who might want to extend this work) should be stated as well. I’m a little skeptical about the broader applicability of this approach, particularly when interventions are not well defined or the treatment variable no longer accords to a binary choice. How does one identify whether a particular neuron is encoding a topic out of 20? What is the intervention in that setting? Or when the authors consider a more nuanced form of grammatical gender? The paper has currently not addressed such scenarios.

Correctness: The method appears to be correct but it requires making certain assumptions that the authors have not stated in the paper. The paper otherwise presents a theoretically sound approach to interpret how neural models encode certain gender (assuming binary) stereotypes.

Clarity: Besides minor suggestions, the paper is well written. - The authors do not define what they mean by bias and do not link the treatment effect to the term. - Figure 3: In one line it says, y_{null}(u) = 0.04/0.22 ~ 0.14. However based on the previous numbers provided by the authors, it should be 0.03/0.22 - Line 110: It its computed -> It is computed - Line 134: I believe this can be phrased better. What do you mean by representation corresponding to the profession word? - Lines 196: It might be worth exploring ways to verify this. - Figure 7(a) is not color-blind friendly. - Lines 305-309 (or something similar) should be front and center in the results section. - Arxiv versions have been cited for a few published peer reviewed papers, I would recommend changing them to reflect the publication venues.

Relation to Prior Work: The paper discusses prior work at length and which makes the paper's contributions clear.

Reproducibility: Yes

Additional Feedback: As certain assumptions are harder to verify in this setting, it would make sense to also check the validity of this approach on multiple architectures as it would provide good supporting evidence should it exist. I’m curious what the authors think about why the findings on the professions dataset do not follow the same trend as on Winobias and Winogender. I would also appreciate the authors acknowledge the unintended harms that could stem from their binarization of gender in the broader impacts section as well as what they intend to do to mitigate such harms.


Review 2

Summary and Contributions: This paper conducted a causal mediation analysis particularly focusing on gender bias application. The analyses show that specific heads, layers, or attentions have effects on those gender-related terms.

Strengths: The paper is very well-written and easy to follow. The examples in Figure 2 and 3 are very informative and straightforward to understand the newly-proposed concepts such as TE and NDE. The proposed approach and its contributions are clearly stated with potential impact and limitations.

Weaknesses: I have a high-level question for authors. What kinds of insights can we learn from this sort of analysis on specific neurons/attentions, beyond the general findings themselves? Are these findings useful for better model development for future research? Are there any takeaways to learn when designing a new model architecture? Or, using the mediation analysis, can we build a bias-free model? It would be much compelling if these motivations are introduced earlier in the paper and provide the potential usefulness of the developed methods. In section 4.2, it is interesting to see some specific heads/layers (i.e., head 10, layer 5) have high attention to the anti-stereotypical candidate, similar to other applications such as coreference resolution. But, then why? Do you have any conjecture about why specific layers/heads have more attention than others? I believe adding more in-depth inspection of the findings makes the paper scientifically strong. The last concern about the paper is small scale of the experiments. The analysis has been conducted on a very specific type; gender and very limited settings such as professions. How does the mediation technique scale to other types of biases or other variants of linguistic phenomena as authors already discussed about that in the paper.

Correctness: Yes. I've checked the methodology and experiments again, and they are all reasonably sound.

Clarity: Yes

Relation to Prior Work: There are some work that uses causal techniques to textual analysis as below: Adapting Text Embeddings for Causal Inference UAI 2020 This might be a personal opinion, but it would be better to read related work right after Introduction especially for those who are not familiar with this field.

Reproducibility: Yes

Additional Feedback:


Review 3

Summary and Contributions: Summary: This work proposes to use causal mediation analysis to interpret which parts of a neural model are causally implicated through a couple intervention approaches. Specifically, the authors focus on pre-trained Transformer language models and investigated which neurons and attention heads might be more attributed to gender bias on a self-adapted dataset focused on a number of professions embedded in sentences with a pre-defined syntactic template. Contributions: 1.     The study demonstrates the feasibility of using casual mediation analysis to analyze gender bias on GPT-2. This paves the way for investigating other bias types (e.g. race/ethnicity, religion) through this approach. 2.     The authors designed intervention methods based on the CMA framework and adapted dataset for measuring biases in professions for the gender-bias probing task.

Strengths: This work is very relevant to the NeurIPS community and is related to several emerging and important areas of research including causal modeling, interpreting pre-trained language models and fairness in NLP/ML. The proposed framework is novel as few have examined these research problems using causal mediation analysis with different types of intervention in the context of neural architectures. This work can potentially lead up to a number of other related efforts.

Weaknesses: 1. Only the reporting clause is examined while the that clause that contains the statement is ignored: In previous bias probing studies, the input content is the entire sentence with the complete context. However, in this paper, only the prompt part (reporting clause) is fed to the language model for analysis. Therefore, the proposed intervention setup effectively only focuses on word level bias probing. 2. Only Nouns are Examined: While it is obvious nouns/professions embody gender bias, verbs could also reveal gender bias. In the templates shown in Figure 8 in the Appendix, the verb “cry” or “drive” could embody implicit bias. However, under the current framework, such potential biases are not investigated.  3. Inter-sentential interactions not considered: For a more comprehensive analysis at the sentence level or beyond, it will require more contextual understanding with more variety of syntactic structures. Therefore, the conclusions drawn in this study that gender bias effects are concentrated in specific components of the model may not generalize well when more complex syntactic and semantic structures and interactions are considered. May I know if the corpus examined contains slightly more complex examples like the following: The nurse cried when he started cooking dinner at home. “nurse”, “cry” and “cook dinner at home” are potentially more likely to be associated with females. How would such a sentence change the probing analysis? How about a sentence like the following? The officer killed the suspect but got away with it due to her political clout. The words/phrases “officer”, “kill”, and “political clout” might have stronger associations with the male gender

Correctness: The method is correct. The claims of the analysis results might not easily generalize well due to the small sample size and the overly simplified setup in the dataset. However, there is limited analysis on “natural direct” as compared to other types of effects. From Figure 5, it is noted that direct effect is a lot more than the indirect effect in the attention heads in Winobias in GPT2-small. Is this similar in other benchmarks and other model variants? How about the direct effects in neurons?

Clarity: The paper is well written with clear explanation of stereotypical and anti-tereotypical concepts through relatable examples. The formulation of causal effect measurements is easy to follow and the experiment results are well presented.

Relation to Prior Work: Prior work is very comprehensive. Section 5.1: The proposed approach bridges both structural and behavioral approaches, analyzing model components through curated examples through a causal framework. Section 5.2: The authors did not explicitly mention how this work is different from other studies that examine gender bias (or other bias).

Reproducibility: Yes

Additional Feedback: The authors could elaborate more on the direct effects in other benchmarks other than the Winobias dataset.


Review 4

Summary and Contributions: The paper propose a new framework to investigate bias in deep language model. The method is grounded in the theory of causal mediation analysis. There measures are presented in the paper namely total effect, direct effect, and indirect effect. Those effects are measured based on a set of designed interventions of the input (total effect and direct effect) or part of the models (indirect effect). Using gender bias as a testbed, the paper shows that gender bias effects are concentrated in specific components of trained language models. The proposed methodology is apt for another type of bias as well.

Strengths: - well motivated for mediation analysis approach in neural language models. - The application of mediation analysis for analyzing attention heads and neurons through interventions make a lot of sense. - well chosen task/datasets to demonstrate the approach. - comprehensive set of experiments and analysis.

Weaknesses: The proposed method is suitable to analyze certain part of the model, however, due to the very large number of parameters in current language model, it is computational demanding and inefficient to analyze the effect of each parameters as well as the interaction between them (for example neuron_i and neuron_j). Nevertheless , this might be an issue for any method that attempt to do so.

Correctness: Yes

Clarity: Yes

Relation to Prior Work: Yes.

Reproducibility: Yes

Additional Feedback: == Post-rebuttal comment == I keep my score and hope to see this paper at the main conference.

[Author Response · NeurIPS 2020]

We thank the reviewers for their thoughtful feedback and helpful suggestions. We address specific points below.

**R1: missing assumptions.** The standard assumptions made in the causality literature are required when we observe only
one outcome per unit and cannot observe the counterfactual outcomes. These assumptions are needed for identification
of causal effects from observed outcomes. However, in our case, we perform two interventions on every unit (and on the
mediators) and observe all counterfactual outcomes. Therefore, these assumptions are not needed for our calculations
of mediation effects, and no statistical bias is expected from the analysis.

**R1: broader applicability.** While we focused in this work on binary interventions and outcomes, the existing literature
on causal mediation analysis enables the study of more general scenarios, including a different combination of variable
types (binary, categorical or continuous) for interventions, mediators and the outcomes.

**R1: defining bias.** Dwork (2012) defines an algorithm to be fair if it gives similar predictions to similar individuals.
The formalization of this definition was extended into Counterfactual Fairness (Kusner, 2017). We will explicitly define
bias as the extent to which an algorithm is not counterfactually fair and draw the connection to our outcome variable $y$.

**R1: dataset differences.** We believe the difference in NIE between the professions dataset (NIE concentrated in initial
layers) and the Winograd-style datasets (NIE concentrated in middle layers) reflects the fact that the former relates to
bias in word embeddings (lexical semantics), while the latter relates to bias in coreference, a higher-level phenomenon.

**R1: harms of gender binarization** We acknowledge that our current discussion of the unintended harms of treating
grammatical gender as binary variable is insufficient. Experimental results on he/they show very similar total effects to
he/she ($\pm15\%$), although with a lower variance. We will add these results and a discussion of the measuring difficulties
of this effect under the singular/plural "they" confounder, as well as suggestions for mitigation, to the main body, and
will extend the impact statement.

**R1/R3: other model variants.** We now have additional results for Tranformer-XL, BERT, DistilBERT, RoBERTa, and
XLNet, which are consistent with the results from GPT-2.

**R2: insights and takeaways.** Debiasing is an important research direction. Although we feel it is beyond the scope of
this paper, we believe our insights point to promising applications in evaluating and developing debiasing techniques.
One could envision manipulating mediators found through our method to reduce gender bias, e.g., setting them to a
null/neutral value. Further study is needed to evaluate how this approach impacts model bias and general performance.

**R2: heads targeting anti-stereotypical candidates.** Attention may capture negative as well as positive relationships,
depending on the head-specific value vectors to which the attention weights are applied. We hypothesize that attention
towards an anti-stereotypical candidate may decrease the probability of it being treated as the antecedent.

**R2: concentration of attention in specific heads.** Past work has shown that attention in middle layers correlates with
coreference (as you allude to), which is tightly related to our analysis of gender bias. Specialization of attention heads
has also been observed more generally, e.g., for various types of dependency relations (Clark et al., 2019). We will
expand the discussion of this point in the camera-ready version.

**R2: other types of biases.** For this novel adaptation of mediation analysis, we perform an extensive analysis of a
specific case study rather than a broader study of multiple phenomena, which would be a great area for future work.

**R2: correctness.** Would R2 point out the methodological problems that warrant a "no" answer to the correctness
question, such that we may address them?

**R2: missing related work.** Thank you for pointing this out. We will add this to the related work.

**R2/R3: limited scale and setup.** We are constrained by available resources. However, we find consistent results across
multiple models/datasets. In addition, the Winograd-style datasets are fairly nuanced in the linguistic phenomena.

**R3: only the reporting clause is reported.** In the professions dataset, we deliberately use verbs that are as neutral as
possible to focus on the profession word and the bias it leads to. In the Winograd-style datasets, the examples are much
more nuanced, in fact containing similar examples to the second one suggested by the reviewer. In this case, the bias
depends on the entire context in the prompt. We agree that our method doesn't take into account potential bias in the
continuation itself.

**R3: only examines nouns.** This is true for the professions dataset, though in the Winograd-style datasets the verbs
play a role as well. We feel that a focused analysis of bias in verbs, while valuable, would warrant a separate study.

**R3: inter-sentential context.** Indeed, we have only looked at intra-sentential context. We note that some contexts are
rather nuanced in these Winograd-style datasets. Our methodology may be applied to inter-sentential contexts as well.

**R3: direct effects.** Figure 5 is representative of the direct effects we observed in all models: direct effects approximate
the difference between the total effects and the indirect effects. We will include additional results on direct effects.

**R3: difference from other studies on gender bias.** The main difference is that our research questions and methodology
focus specifically on mediators in bias by performing interventions. We will clarify this in the related work.

**R4: computational cost.** We recognize that that computational cost is non-trivial. We discuss computational complexity
in Appendix D with respect to the subset selection algorithm, but we will also discuss more generally in the main body.

[Meta-Review · NeurIPS 2020]

The paper studies the problem of bias in neural models where the proposed solution is based on causal mediation analysis. The focus of the paper is on pre-trained transformer language models, GPT-2. The proposed method of using mediation analysis for analyzing attention heads and neurons through interventions is novel and interesting, and can be generalized to other types of biases. The paper is well-written, and experiments are thorough.